# Thrombectomy of an Acute Ischemic Stroke in a Child with COVID-19 and MIS-C: Case Analysis and Literature Context

**DOI:** 10.3390/children10050851

**Published:** 2023-05-09

**Authors:** Paul R. Carney, Dakota W. Stevenson, Edith Riggs, Matilda Dervisevic, Constanza X. Carney, Camilo R. Gomez

**Affiliations:** 1Department of Child Health, University of Missouri School of Medicine, Columbia, MO 65201, USA; 2Department of Neurology, University of Missouri School of Medicine, Colombia, MO 65212, USA; dwsrcr@health.missouri.edu (D.W.S.);; 3Department of Epidemiology, George Washington University, Washington, DC 20052, USA

**Keywords:** stroke, children, COVID-19, MIS-C, thrombectomy

## Abstract

We describe a very young child who developed an acute ischemic stroke from a LAO, while affected by COVID-19 and MIS-C, and whom we treated successfully with thrombectomy. We compare his clinical and imaging findings with those of the existing case reports, and we explore the multifactorial nature of such a neurovascular complication, particularly in the context of the most recent publications regarding the multifactorial endothelial derangements produced by the illness.

## 1. Introduction

While most children with coronavirus disease 2019 (COVID-19) are asymptomatic or experience only mild symptoms that seldom require hospitalization, multisystem inflammatory syndrome in children (MIS-C) is a rare but serious post-infectious inflammatory condition, with a “Kawasaki-like” course and multiorgan dysfunction that can rapidly complicate the progression of the original illness [1,2,3,4,5,6]. Acute COVID-19 infection in adults has been associated with high rates of thromboembolic events, including ischemic stroke, presumably due to COVID-19-induced thrombophilia and endothelial dysfunction [7]. However, although children exhibiting MIS-C have been found to display abnormal coagulation markers, it is unclear whether the condition is associated with increased rates of thromboembolism or not, and the administration of anticoagulants to affected patients varies between institutions [4,5]. Amidst such uncertainty, a recently published systematic review, reporting on over 600 children with MIS-C, did not contain a single case of ischemic stroke as a complicating incident, despite 10–20% of the reported patients displaying non-specific neurologic symptoms [1]. Likewise, another case series from 52 different institutions, including over 1600 children and adolescents hospitalized for COVID-19 or MIS-C, only contained 12 (0.7%) cases of complicating stroke [8]. At the time of this writing, there have been several reports of pediatric patients suffering ischemic stroke from large arterial occlusion (LAO) in the context of COVID-19 [8,9,10]. However, most of these patients were older children and adolescents who were treated conservatively, and the case reports lack uniform imaging confirmation. We describe a very young child who developed an acute ischemic stroke from a LAO, while affected by COVID-19 and MIS-C, and whom we treated successfully with thrombectomy. We compare his clinical and imaging findings with those of the existing case reports, and we explore the multifactorial nature of such a neurovascular complication.

## 2. Case Presentation

A previously healthy 3-year-old boy was admitted to the University of Missouri Columbia Children’s Hospital on 16 December 2020, with a 4-day history of acute gastrointestinal symptoms (i.e., nausea, vomiting, abdominal pain), fever, rash, shock, mild left ventricular myocardial dysfunction, elevated inflammatory markers (C-reactive protein, procalcitonin, ferritin, transaminases, interleukin-1, interleukin-6), and evidence of COVID-19 (RT-PCR, antigen test, serology positive) (Table 1), thereby satisfying the criteria of MIS-C [1,3,4,5,6]. He was rapidly treated with intravenous methylprednisolone (2 mg/kg/day, divided every 6 h), intravenous immunoglobulin (2 g/kg), enteral aspirin (81 mg per day), and prophylactic subcutaneous low-molecular-weight heparinoid (0.5 mg/kg twice daily). On the second inpatient day, he developed sudden weakness in the right arm and leg, aphasia, and an overall neurologic deficit that amounted to a Pediatric National Institutes Stroke Scale (PedNIHSS) score of 26. Magnetic resonance imaging (MRI) showed evidence of restricted diffusion consistent with ischemia of the territory of the middle division of the left middle cerebral artery—LMCA (md)—(Figure 1a), and yet, minimally increased signal of the analogous volume of tissue in the Flow Attenuation Inversion Recovery (FLAIR) sequence (Figure 1b). In addition, magnetic resonance angiography (MRA) demonstrated occlusion of the LMCA (md) (Figure 1c), leading to the decision to proceed with urgent thrombectomy. Cerebral angiography confirmed the expected occluded artery (Figure 2a), which was successfully recanalized by retriever thrombectomy (Figure 2b,c). Successful reperfusion [Thrombolysis in Cerebral Infarction (TICI) = 2b] of the affected vascular territory was immediately confirmed via digital parenchymography (Figure 3). Over the 12–24 h following the procedure, the child regained spontaneous movement of his right arm and leg and began to communicate verbally (PedNIHSS = 10). He continued to progressively improve, reaching near-normal motor and language function (PedNIHSS = 5) at hospital discharge one week later, and displayed no neurologic deficits at 30 days in the outpatient clinic. Additional imaging studies, including echocardiography and venous Doppler ultrasound of all limbs, proved unremarkable. However, he was found to be heterozygous for Factor V Leiden mutation.

## 3. Discussion

The neurologic complications in COVID-19 patients constitute an evolving area of study [11,12]. However, acute ischemic stroke remains an infrequent complication even in pediatric patients, having been reported in less than 1% of cases [13,14]. More relevant to our case, targeted surveillance of the pediatric manifestations of MIS-C across 26 states showed neurologic involvement in only 5% of children and 11% of adolescents [3].

Tiwari et al. reported the first case of a nine-year-old girl who presented with ischemic stroke while suffering an active COVID-19 infection complicated by MIS-C [15]. Despite aggressive management, the child died, having suffered extensive and multifocal cerebral infarctions [15]. In addition, her computed tomography angiogram (CTA) demonstrated multifocal stenotic lesions in the major arterial branches of the carotid system [15]. Previously, Beslow et al. had described eight pediatric patients with COVID-19 and ischemic stroke, their ages ranging between four days and fourteen years old, and only two of whom were suspected of having concurrent MIS-C [13]. However, these two cases were confounded by one having sickle cell disease and the other suffering the stroke while undergoing extracorporeal membrane oxygenation (ECMO), both risk factors for stroke in their own right [16,17]. A prevailing notion is that children with ischemic stroke associated with COVID-19, irrespective of whether they meet the criteria for MIS-C or not, may be afflicted by a large arterial inflammatory vasculopathy (i.e., arteritis) that predisposes them to the development of cerebral ischemia and stroke [7,9,18,19]. At the time of this writing, we found only four previously reported cases of ischemic stroke caused by a LAO in the context of COVID-19, all considerably older than our case, and only two of which were also found to have MIS-C (Table 2). Two of these patients were found to have imaging evidence of a cerebral arteriopathy, while the other two displayed left ventricular dysfunction that suggested a cardiogenic embolism. Only one was treated endovascularly (i.e., retriever thrombectomy), but suffered re-occlusion and ultimately suffered a disabling stroke [9].

The association of COVID-19 infection with thrombophilia and cardiovascular urgencies, including ischemic stroke, is well known in adults but has only recently been given its due attention in the pediatric literature [13,20]. Our patient, the fifth identified case (Table 2), presents a unique set of attributes, including the fact he is the youngest subject reported with ischemic stroke from a LAO in the context of COVID-19, and treated by thrombectomy. Considering our current knowledge about COVID-19 and MIS-C, it is therefore intriguing to examine how all the concurrent features of this case may have contributed to his acute LAO, including the potential interactions of (a) COVID-19 thrombophilia, (b) factor V Leiden heterozygosity, (c) MIS-C cerebral arteritis, and (d) systemic hemodynamic instability. A theoretical framework, in which to consider these interactions, is illustrated in Figure 4, capitalizing on the concepts elegantly exposed by Libby and Lüscher [7].

As with any other analysis of ischemic stroke, we have historically considered three pathologic domains that interact as precursors for the development of acute cerebral arterial embolism: cerebrovascular (i.e., “vessels”), cardiac (i.e., “heart”), and hematologic (i.e., “blood”). These domains, operating interdependently, constitute a construct that can be greatly affected by the numerous derangements of endothelial function identified in patients with COVID-19 and MIS-C [7]. As such, endothelial inflammatory changes directly affecting the cerebral arteries become the pathologic substrate of the arteritis previously identified in these children, and which can be readily imaged using MRI-based vessel wall imaging techniques [9,18,19]. In addition, direct procoagulant abnormalities of endothelial function, as well as primary thrombophilias, are bound to add to the problem by promoting intravascular in situ thrombosis [21,22]. Such development of intravascular thrombi may directly occlude a cerebral artery, or else create a nidus for embolism. The latter could easily occur within the left ventricle of a failing heart affected by myocarditis (i.e., “cardiogenic” embolism), or even in the deep venous system of the pelvis from where it can travel, via naturally existing right-to-left shunts, to the cerebral circulation (i.e., “paradoxical” embolism via a patent foramen ovale).

In the context of these concepts, our case underscores the importance of (a) rapidly identifying neurologic changes in these patients, recognizing the possibility of acute cerebral ischemia as the root cause of such changes, (b) the critical role of urgent non-invasive cerebrovascular imaging (i.e., MRI and MRA) in selecting patients who are likely to benefit from urgent thrombectomy, and (c) the realization that thrombectomy is only the “tip of the spear”, and that timely and aggressive therapy with immune modulators (e.g., steroids, immunoglobulin), judicious anticoagulation, and comprehensive hemodynamic support are crucial for reaching favorable functional outcomes in children with MIS-C-related acute ischemic stroke [4,5]. Further studies are required to understand the additional cerebrovascular risk conferred by MIS-C to children with COVID-19 infection. In the interim, we cannot but recommend a high index of suspicion, a keen sense of situational awareness, liberal utilization of MRI and MRA, and close communication with experienced neurointerventional teams.

## Figures and Tables

**Figure 1 children-10-00851-f001:**
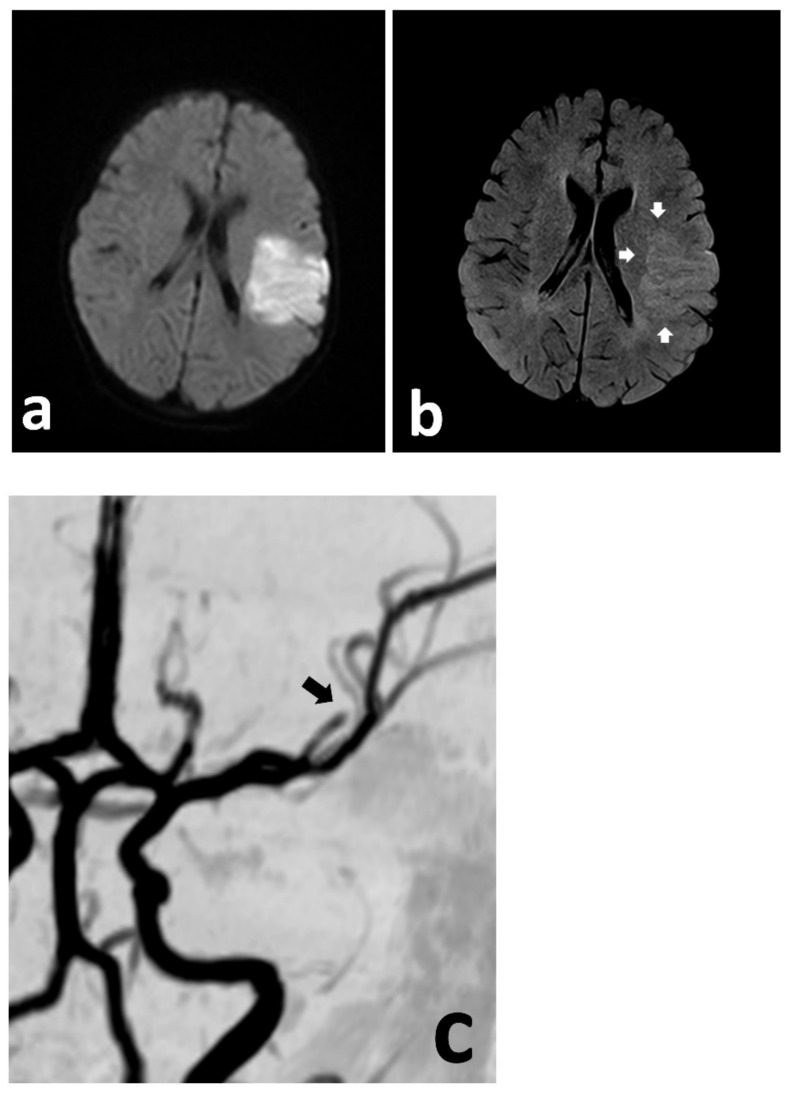
Urgent MRI study demonstrating focally restricted diffusion in the MCA (md) territory (**a**), with only discrete hyperintensity in the FLAIR sequence (**b**) (white arrows), indicating at least a partial mismatch. The MRA demonstrates cut-off of the MCA (md) (**c**) (black arrow).

**Figure 2 children-10-00851-f002:**
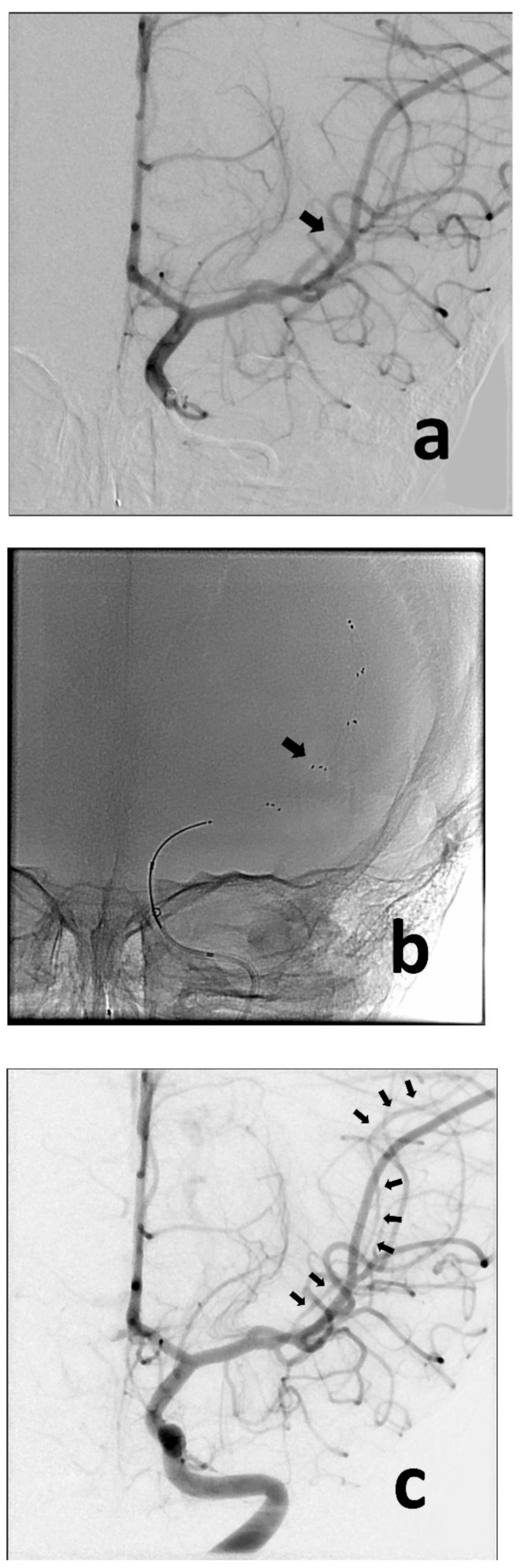
Anteroposterior angiographic and fluoroscopic views of the thrombectomy procedure. Baseline image confirms occlusion of the MCA (md) (**a**) (black arrow). The retriever was deployed across the point of occlusion (i.e., normal-to-normal) (**b**) (black arrow). Following removal of the retriever, the MCA (md) is now visualized (**c**) (multiple black arrows), despite showing some spasm (typically caused by the device).

**Figure 3 children-10-00851-f003:**
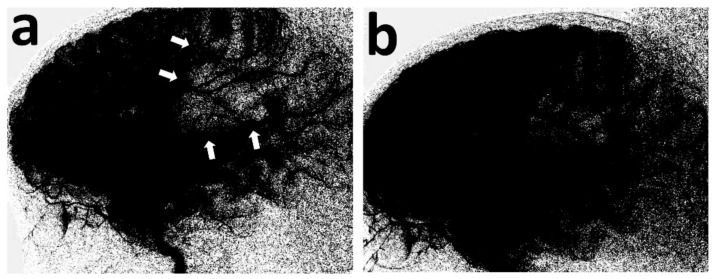
Lateral view of the digital parenchymography, before and after thrombectomy. At baseline, it demonstrates the lack of opacification of the MCA (md) territory (**a**) (white arrows), which materially opacifies, following removal of the retriever (**b**).

**Figure 4 children-10-00851-f004:**
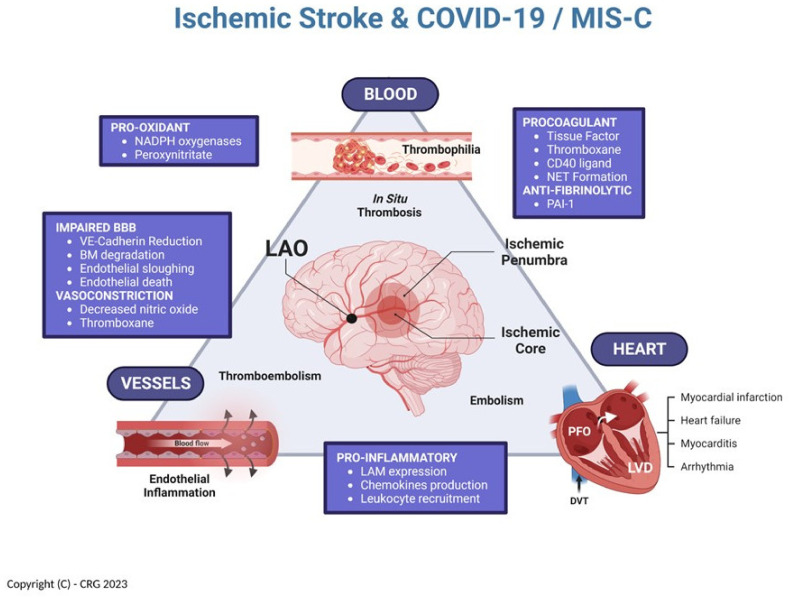
Multifactorial relationship between COVID-19, MIS-C, and ischemic stroke [inspired by Libby and Lüscher (7)]. NADPH = reduced nicotinamide adenine dinucleotide phosphate; NET = neutrophil extracellular traps; PAI-1 = plasminogen activator inhibitor 1; LAM = leukocytes adhesion molecules; VE = vascular endothelial; BM = basement membrane; LVD = left ventricular dysfunction; PFO = patent foramen ovale; DVT = deep venous thrombosis. (Created in Biorender.com).

**Table 1 children-10-00851-t001:** Results of the admission laboratory studies. WBC = white blood cell count; AST-SGOT = aspartate transferase-glutamic-oxaloacetic transaminase; NP = natriuretic peptide; proBNP = N-terminal pro-b-type natriuretic peptide; CRP = C-reactive protein; SARS-CoV-2 = severe acute respiratory syndrome; PCR = polymerase chain reaction; CoV-2 = coronavirus; IgM = immunoglobulin M; IgG = immunoglobulin G; MTHFR = methylenetetrahydrofolate reductase; IL = interleukin; TNF = tumor necrosis factor,.

TEST	RESULT	REFERENCE
**HEMATOLOGY**		
WBC (neutrophils per mm^3^)	22.56 (18.26)	4–12
Absolute lymphocytes (per L)	1.9 × 10^9^	1.50–7.00
Platelet count—per mm^3^	442	150–450
**SERUM CHEMISTRY**		
AST-SGOT (units per L)	50	<40
NT-proBNP (pg/mL)	36,163	0.00–125.00
Ferritin (ng/mL)	3736	30.0–400.0
Triglyceride (mg/dL)	222	0–75
**INFLAMMATORY MARKERS**		
CRP (mg/dL)	16.96	0.00–0.50
Procalcitonin (ng/mL)	12.40	0.00–0.05
SARS-CoV-2 PCR	Detected	
SARS-CoV-2 Ab, IgM/IgG	Reactive	
Influenza A and B rapid	Negative	
Blood culture at 96 h	No growth	
IL 2 receptor (pg/mL)	23,183.8	175.3–958.2
IL 10 (pg/mL)	228.1	≤2.1
IL 13 (pg/mL)	13.9	≤2.3
IL 17 (pg/mL)	4.6	≤1.4
IL 8 (pg/mL)	11.7	≤2.0
IL 6 (pg/mL)	41.9	≤3.0
TNF alpha (pg/mL)	12.5	≤7.2
**HEMOSTASIS**		
Prothrombin Time (s)	14.7	12.4–14.4
d-Dimer (mcg/mL)	3.81	0.00–0.50
**THROMBOPHILIA PANEL**		
Factor V Leiden mutation	Heterozygous	
MTHFR mutation	Not detected	
PT 202101G>A mutation	Not detected	
AT III	96 units/dL	
Protein C activity	137%	
Protein S activity	71%	

**Table 2 children-10-00851-t002:** Pediatric patients confirmed to have suffered ischemic stroke from LAO. LAO = large arterial occlusion; NIHSS = National Institutes of Health Stroke Scale; LMCA = left middle cerebral artery; LICA = left internal carotid artery; id = inferior division; md = middle division; t = terminus; Endo = endovascular; Rx = treatment; MIS-C = multisystem inflammatory syndrome in children; Dx = diagnosis; LVD = left ventricular dysfunction; LVT = left ventricular thrombus; mRS = modified Rankin score.

Patient [Source]	Age (Years)	Sex	NIHSS	LAO	Endo Rx	MIS-C	Causative Dx	Outcome
1 [9]	8	Girl	15	LMCA	Yes	No	Arteritis	Ambulating with walker and minimal assistance around day 30 (mRS = 4)
2 [9]	16	Boy	19	LMCA	No	No	Arteritis	Aphasia and right hemiparesis around day 60 (mRS = 3)
3 [10]	15	Girl	15	LMCA (pd)	No	Yes	LVD	No deficit at 30 days
4 [10]	16	Girl	16	LICA (t)	No	Yes	LVT	Independent but aphasic at 41 days
5 [Ours]	3	Boy		LMCA (md)	Yes	Yes	Factor V Leiden	Essentially normal at 30 days

## Data Availability

Not Applicable.

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
