# Peer review of "Thrombectomy of an Acute Ischemic Stroke in a Child with COVID-19 and MIS-C: Case Analysis and Literature Context"

_children, 2023, doi:10.3390/children10050851_

Round 1

Reviewer 1 Report

The manuscript describes a case of acute ischemic stroke in a 3-year-old child associated with COVID-19. So far, there are no reports of this clinical manifestation in children being treated by thrombectomy. Thus, the study is important in the description of this rare presentation of the disease, especially in children. Below are some suggestions/corrections for the authors:

1- In the title of Table 1, I suggest including “Clinical characteristics and laboratory markers of a Young Child Presenting with Ischemic Stroke”

2- In the first paragraph of the case presentation topic, the authors report that the child had evidence for COVID-19 (RT-PCR, antigen test, positive serology), however, these RT-qPCR and antigen test results are not presented in Table 1. For example, if the RT-qPCR or antigen test result is positive, it would not be evidence but confirmation of SARS-CoV-2. These data need to be clear in Table 1

3- Where and when was the child admitted to start the diagnosis and treatment? How long did the hospitalization last?

4- The presence of antibodies (IgM and/or IgG) may indicate the child's previous contact with the virus. Is there information if this child previously had COVID-19 or was vaccinated?

5- I suggest a complete review of Table 1, identifying in the legend the meaning of all the acronyms present in the Table. Many acronyms (For example, PT, CRP, AST-SGOT) do not show their meaning in the legend.

6- In the caption, the acronym PCR: polymerase chain reaction appears, however, this acronym does not appear in the table. The acronym "IT" appears in the legend as Interleukin, but it should be “IL”

7- In the table, after IL-17, only the acronym IL appears, however, it does not inform which interleukin would be. What is the Interleukin presented in this case?

8- In addition to the neutrophil count, the authors could inform the lymphocyte count.

9- I suggest placing the reference values for all serological laboratory markers directly in the table. This information needs to be clearer in Table 1. In addition, it is necessary to standardize the presentation of these values.

Author Response

1. Table 1 title has been changed to Clinical characteristics and Laboratory Markers of a Young Child Presenting with Ischemic Stroke.
2. RT-qPCR and antigen test results are included in Table 1.
3. Detailed regarding patient admission and dates are included in the case summary.
4. The child was not vaccinated and did not have prior COVID-19 infection information is now included in the case summary.
5. Table 1 legend includes the meaning of all acronyms.
6. IT is denoted interleukin in Table 1 as suggested.
7. The lymphocyte count is included in Table 1.
8. Reference values for all serological values are included in Table 1.

Reviewer 2 Report

Thank you for the opportunity to review this very interesting and clearly rare presentation of large vessel AIS in a child with ongoing Covid-related MISC. The case is presented extremely well and in logical fashion, with helpful references to laboratory and imaging study results.

May I suggest a few recent references to strengthen the introduction and discussion sections with respect to the incidence, type, and potential pathophysiology of AIS this population. In your introductory paragraph you state that this presentation (large vessel AIS in the setting of MISC and mild shock) has not been previously reported in children:

(1) Chang J et al. [Pediatr Neurol 2022;126:104-7] reports on two adolescent females presenting with MISC who develop AIS. The authors also discuss the Beslow et al study you reference, pointing out that only 8/971 pediatric patients developed AIS, 4 of which were LVOs. The authors also propose an underlying pathophysiology.

(2) Velleux MJ et al [Pediatr Neurol In Press, https://doi.org/10.1016/j.pediatrneurol.2022.10.003] reports on 16 pediatric patients with AIS, 3 diagnosed with MISC. At their institution the peak stroke incidence was February 2021, 2 months following the peak Covid infection incidence, and 1 month following the peak MISC incidence. LVO was most common (9/16, 56%), all involving the MCA territory. The authors compare the cases to historical AIS patients are their institution, and theorize that AIS could be a presentation of the prothrombotic state induced by post-Covid hyperinflammation, but also speculate that the risk for AIS is distinct from the risk for MISC.

(3) Beslow LA et al [Stroke 2022;53:2497-2503] identified 335 cases of AIS in children, with 23 (6.9%) positive for Covid. They determined Covid to be the primary risk factor for AIS in 6/23, contributory in 13 (10 of which had subclinical Covid infection), and incidental in 3. The authors speculate about the underlying pathophysiology in children, pointing to complement activation and resulting thrombosis and endothelial activation.

Author Response

1. Suggested references when publicly available are now included in the manuscript
introduction.

Round 2

Reviewer 2 Report

Thank you very much for addressing my earlier comments.